# Measuring Oral Reading Fluency (ORF) Computer-Based and Paper-Based: Examining the Mode Effect in Reading Accuracy and Reading Fluency

Jana Jungjohann [1,*], Jeffrey M. DeVries [2] and Markus Gebhardt [1]

1 Faculty of Human Science, University of Regensburg, 93053 Regensburg, Germany; markus.gebhardt@paedagogik.uni-regensburg.de
2 Center for Research on Education and School Development, TU Dortmund University, 44227 Dortmund, Germany; jeffrey.devries@tu-dortmund.de
* Correspondence: jana.jungjohann@ur.de

**Abstract:** Internationally, teachers use oral reading fluency (ORF) measurements to monitor learning progress in reading and adapt instruction to the individual needs of students. In ORF measures, the child reads aloud single syllables, words, or short passages, and the teacher rates in parallel at which items the child makes a mistake. Since administering paper-based ORF requires increased effort on the part of teachers, computer-based test administration is available. However, there are still concerns about the comparability of paper-based and computer-based test modes. In our study, we examine mode effects between paper-based and computer-based test scores for both reading speed and reading accuracy using a German-language ORF assessment for progress monitoring. 2nd- and 3rd-year-students ($N = 359$) with and without special education needs participated in the study. Results show comparable and high reliability ($r > 0.76$) and no differential item functioning for both test modes. Furthermore, students showed significantly higher reading speed on the paper-based test, while no differences were found in reading accuracy. In the absence of differential item functioning, we discuss how mean differences can be accounted for, how teachers can be trained to use the different test modes, and how computer-based tests can be safeguarded in practice.

**Keywords:** computer-based assessment; differential item functioning; oral reading fluency; mode effects; paper-based assessment; progress monitoring

## 1. Introduction

Classroom-based assessment data are used in multi-tiered systems of support such as response-to-intervention models (RTI) by teachers to shape teaching and individual learning in the sense of data-based decision-making (DBDM) [1], with the goal of improving the learning environment for all students in inclusive, special, and regular schools [2–4]. Not all students benefit in the same way from regular instruction (e.g., [5]), and therefore, individualized instructions are needed to meet individual learning needs [6]. In particular, students with special education needs (SENs) are at high risk for difficulties in reading and mathematics [7,8]. RTI is a framework in which multiple levels of support are implemented at different intensities to respond to students' individual learning needs. The main idea is that the greater the student's need, the greater the intensity of the intervention [9]. Typically, RTI models are structured with three tiers. In the first tier, teachers design high-quality instruction for the entire learning group and assess students' response to the instruction three times per school year. In the second tier, students who have not sufficiently benefited from the first tier's efforts receive more intensive and preventive interventions in small groups. Brief interventions are usually provided daily and evaluated monthly. At the third tier students with severe difficulties receive intensive and individualized evidence-based interventions, the success of which may be evaluated as often as weekly. To evaluate

students' responses to interventions, classroom-based assessments are implemented at all tiers. Classroom-based assessments include both standardized summative and formative assessments [10]. Especially for students with learning difficulties and SENs in inclusive education, the use of formative assessments (i.e., 'assessments for learning', [11]) has a positive impact on their learning development (e.g., [12–14]). Formative assessments include repeated measures of specific learning outcomes that allow the process of learning to be visualized [15]. Such assessments are considered crucial components of inclusive education because individual learning needs are expected to be broader than in regular classrooms due to higher heterogeneity in student achievement [16]. Nevertheless, teachers' familiarity with formative assessments in inclusive education practice is currently still lower than that with summative assessments [10,17]. A well-known form of formative assessment is curriculum-based measurement (CBM) [18], which was designed to solve academic difficulties in special education. Multiple instruments have been supported in research as reliable and valid [19–21].

Curriculum-based measurements are used for early identification of students who are at high risk for serious learning difficulties. CBM entails easy-to-use tests with a short administration time of a few minutes that can be used during lessons. The tests are cross-sectionally used with approximately three to four measurement points per school year as a screening tool to identify non-responders or longitudinally with up to weekly measurements to evaluate the success of support measures [22]. Therefore, CBM tests offer multiple parallel versions with comparable difficulty so that the tests can be repeated. The underlying idea is that over time, test results can be visualized in a graph. With the help of the learning graph and information about the current instruction, teachers can then make decisions about whether to continue or adapt the instruction in case of a lack of progress. CBM is available for different learning areas in reading, spelling, writing, or mathematics. The tests differ in whether they are constructed using the robust indicator approach or curriculum sampling [23]. With one representative type of task, robust indicator tests measure a broad skill that correlates highly with the end-of-a-school-year achievement. For example, in the area of literacy, this holds for oral reading fluency (ORF) tasks [24,25] or reading comprehension by maze tasks [26]. In most ORF tests, the number of correctly read syllables or words within 60 s is scored to quantify the reading rate in terms of reading speed [27]. Tests using the curriculum sampling approach include different types of tasks that measure multiple subskills at different levels of difficulty. This approach is typically chosen for CBM measuring mathematical skills or competencies [28]. In general, CBM tests are constructed as speeded tests with a fixed time limit [29,30]. This means that students can complete as many tasks as they can within the fixed test time. The number of correctly solved tasks then forms the total sum score, and the percentage correct can be interpreted as reading accuracy for educational purposes. Therefore, theory-based item design and empirical validation using item response theory (IRT) approaches are recommended [31].

Digital devices such as computers and tablets, as well as WIFI access, are increasingly available in classrooms. These technologies are frequently used to administer computer-based or web-based CBM (e.g., [32–35]). In many cases and due to Coronavirus Disease 2019 (COVID-19), computer-based assessments have replaced paper-based assessments because mode equivalence including test fairness across different student groups (e.g., gender, migration background, and SEN) has been assumed. A particular argument in favor of the use of computer-based assessments is that students like this test mode better than paper-based assessments [36–38]. At the same time, there are still teachers who have a negative attitude towards computer-based assessments and prefer to use paper-based assessments in class [39], who demand more training in the use of computer-based assessments or who criticize a lack of technical equipment in schools [40,41]. As a result, both computer-based and paper-based assessments are used in parallel in school practice. In the sense of mode effects [42], the question arises if student performance in CBM differs based on the test mode.

### 1.1. Test Mode and CBM

With the advent of computers in classrooms, the first approaches to administer CBM digitally instead of on paper emerged [43]. The design of computer-based CBM aims to make administration simple and standardized to keep scoring as quick as possible without compromising the usefulness for teachers [44–46]. For practitioners, computer-based tests take over administrative and organizational tasks and can provide automated scoring of test results alongside additional support for interpretation (e.g., the presence of a trend line). Web-based assessments also have the advantage that they do not require installation on a local device. They can be used on any device with an internet connection, and the students' results are available to teachers independently of the school computer. Proponents of the paper-based test mode emphasize that the execution of the assessment depends on the availability and functionality of digital devices. In research, digital test procedures are used to increase the reliability of the tests via further item parameters such as processing speed per item (e.g., [31,47]) or to support teachers in decision-making (e.g., [10,48]).

ORF tasks differ in test administration compared to other CBM tests. They are administered as individual tests because a person with good reading skills needs to be measured in reading fluency. Reading fluency represents an interplay of accuracy, speed, and prosodic aspects and is understood as the ability to read texts automatically and unconsciously [49]. For early reading learners, it is considered as an important learning step toward becoming a proficient reader [50] because it affects comprehension to a significant degree [51]. Without sufficient reading fluency, readers focus on decoding which limits their ability to comprehend at the same time. Reading aloud is considered as a robust indicator of reading proficiency [25,27] and is measured via ORF tasks.

In ORF tasks, students are presented with either connected texts or lists of unconnected syllables, words, or pseudowords. They are asked to correctly read as many items (i.e., syllables or words) as possible in 60 s. The number of items read correctly per minute is often also called the sum score in ORF tasks. Different types of items are used in ORF tasks because the sum score is sensitive to the difficulty level of the items [20]. Unconnected word lists with single syllables or simple words are particularly suitable for measuring the reading performance of students with low skills such as students at the beginning of reading acquisition or those with special education needs [34,52].

### 1.2. Mode Effects on CBM

To date, there have been few empirical studies examining differences in equivalence between computer-based and paper-based CBM for students. Predominantly, studies examine test formats in which students complete tasks in the digital and analog forms independently and in groups (i.e., CBM maze or math CBM). Many studies report significantly lower sum scores in the digital tests while concluding that the test formats are not comparable [53–57]. In addition, Støle et al. [57] report a greater disadvantage for girls with high reading skills in digital test administration. Blumenthal and Blumenthal [36] found no differences in sum scores but showed differences at the item level. They compared CBM to measure computation skill (i.e., addition and subtraction) with 98 fourth-grade students using IRT analysis. Item analyses showed that students were more likely to solve the tasks correctly on paper than on the tablet. In the case of a math CBM, it is conceivable that students may have made notes for side calculations on the paper-based assessment, providing additional assurance of their results.

Regarding ORF tasks, no mode effect studies could be found by the authors in peer-reviewed journals. In her dissertation, Schaffer Seits [58] examined the mode effects in ORF based on whether the reading texts were shown to students on paper or computer screen. For this purpose, 108 students from second to fifth grade each completed two comparable ORF tasks in random order; one passage was read on paper, and the other was read on a computer screen. Measured by the sum score, the students read significantly more words correctly in one minute on paper compared to on screen. However, in both test modes, reading errors were documented manually by the test giver. This means for

both computer- and paper-based tests that the scoring methods were identical. Mode effect studies of summative reading assessments support Schaffer Seits' [58] assumption that students read faster on paper and extend this to conclude that reading accuracy is lower on the computer [59].

## 2. Present Study

The literature review reveals a fundamental research desideratum in the area of mode effect studies with CBM, which is particularly serious in ORF. The authors could not find any research examining the reading speed (i.e., sum scores) and reading accuracy (i.e., percentage correct) of ORF assessments in which students read computer-based on the screen and teachers use the technology for assessment. This gap is closed by the present study. Using IRT analyses, two similar ORF measures (i.e., one paper-based and one computer-based test administration) are compared and examined for differences in reading accuracy and reading rate. Therefore, we ask:

1. Does the computer-based ORF test administration have a similar reliability as the paper-based one?
2. Are there differences in the sum scores of computer-based and paper-based ORF procedures?
3. Are there differences in the percentage correct of computer-based and paper-based ORF procedures?
4. Are there differences in item functioning across test modes and student background characteristics?

### 2.1. Participants

Participants ($n = 359$) were 2nd- and 3rd-year students in regular schools in western Germany. They were recruited from 19 classrooms in eight schools. Table 1 provides a detailed overview of the sample concerning grade level, gender, migration background, and SEN status. Instructors indicated whether each child had a migration background and/or SENs. SENs were grouped across several categories, including learning disorders ($n = 14$), language needs ($n = 12$), cognitive developmental issues ($n = 1$), and others ($n = 26$). Written consent from parents was obtained for all participants, and participation was voluntary. Students could withdraw from participation at any time without providing a reason.

**Table 1.** Sample description.

|  | Count (*N*) | Percent |
|---|---|---|
| Total Sample | 359 | 100.0% |
| With Migration background | 227 | 63.2% |
| With Special Education Needs | 53 | 14.7% |
| Male | 198 | 55.2% |
| 3rd Year | 193 | 53.8% |

### 2.2. Instrument

Two equivalent parallel versions of an ORF assessment for German primary school students [60] were used. The ORF is structured as a word list and contains single syllables as items. Thus, pure reading synthesis is tested without segmentation of syllables and retrieval from the mental lexicon [61]. All syllables are generated according to defined rules for their respective difficulty level. The syllables contain only selected consonants and all vowels. Both open (ending with a vowel) and closed (ending with a consonant) syllables are allowed. This CBM form is especially appropriate for beginning readers and students with reading difficulties [53]. Each of the parallel forms contains 114 syllables. Students have 60 s to correctly read aloud as many syllables as they can. An adult with high reading skills evaluates during the read-aloud which syllables were read correctly and which were read incorrectly. When the 60 s time limit has expired, the platform automatically ends the

measurement. The syllable that is currently displayed when the test time is running out may be completed by the child until the end. Previous research has tested the psychometric quality of the ORF tests according to item response theory (IRT) and the test–retest reliability of the used ORF test is $r = 0.76$ [34]. In addition, the ORF test used shows moderate to high correlations ($r = 0.65$–$0.71$) with the standardized reading comprehension test ELFE II [62] at the word, sentence, and text levels [12].

This ORF is implemented as a computer-based assessment in an online platform for progress monitoring (levumi.de, [63]). The computer-based version is performed on a computer with a wireless internet connection. In this test mode, the platform monitors the test time while the teacher evaluates the reading errors and informs the platform via the keyboard if the item was solved correctly. The platform provides an automated scoring of the sum score. In addition, the platform provides feedback to each child after each measure in terms of an individual reference norm [64]. The platform compares the number of correctly solved tasks (here: correctly read syllables within 60 s) and compares this sum score with the sum score of the previous measure. Depending on whether the difference is positive or negative, the mascot motivates the child to practice more or reinforces their learning success. The paper-based form is a printable worksheet placed in front of the student at the desk. It is the teacher's job to stop the test time and document the correct and incorrect results. Feedback can be provided by the teacher after the sum score has been evaluated manually using a template.

### 2.3. Procedure

Participants were given both the computer- and paper-based versions of the ORF test. They were randomly given either the paper- or the computer-based test first. All students received feedback on their sum scores after the administration of both tests. Data were collected by a trained research assistant.

### 2.4. Measures

Analyses are based on the sum score, the percentage correct, and the modeled ability levels via IRT (see below). The sum score is the total number of correctly answered syllables and reflected reading speed. Both the percentage correct and the modeled ability levels (theta in the IRT analyses, see below) reflect accuracy because they account for correct responses based on problem difficulty. The online platform for progress monitoring levumi.de provided both sum scores and percentages correct values automatically. Thus, no errors in calculation are expected. The values of the paper-based version were initially obtained by research assistants using a standardized template. All authors accompanied and verified this process. Unreached syllables were considered missing and not incorrect, in the IRT analyses.

### 2.5. Analyses

The data and syntax for all IRT analyses are available under https://osf.io/s4ruc/ (accessed on 12 June 2023).

A one-dimensional IRT model (i.e., a 1PL or Rasch model) was applied to both computer and paper-based data. A combined group-based Rasch model was calculated with the test modality as a grouping variable using the module "TAM" in R [65,66]. Separate similar models were calculated using only the data from the computer-based and paper-based test forms. Syllables that were not reached were considered missing and not incorrect. We compared performance (sum scores and percentage correct) via a within-groups *t*-test in the computer- and paper-based form.

Differential item functioning for test modality, SEN, gender, and migration background was tested via Mantel–Haenszel statistics [67] using the module "sirt" [68]. A Bonferroni correction was applied based on the number of syllables tested.

After examining the possibility of DIF based on test modality, we examined the number of syllables successfully completed via a within groups comparison. We examined both the

number of syllables completed correctly (i.e., sum score) and the IRT estimate of person ability (i.e., theta value). We further compared performance across the comparison groups based on gender, SEN, grade level, and migration background.

To test the possible impact of gender, migration background, the presence of SEN, and grade level on performance, we conducted a two-level regression with group mean centering at the classroom level, with gender, migration background, and SEN as within variables and school year as a between variable. This analysis was performed using Mplus 8.6 [69].

## 3. Results

### 3.1. Model Fits

Single-point estimates of reliability showed that the test was reliable based on WLE (Warm's Likelihood Estimate) and EAP (expected a posterior), with $WLE_{computer-based}$ reliability = 0.763, $WLE_{paper-based}$ reliability = 0.796; $EAP_{computer-based}$ reliability = 0.784, $EAP_{paper-based}$ reliability = 0.793. A combined model also had high reliability, $WLE_{combined}$ reliability = 0.772; $EAP_{combined}$ reliability = 0.784. Figure 1 shows that when all syllables are answered, the test is generally well-targeted for participants of low through moderate ability. Further, the information curves are quite consistent between test modalities. It is important to note that when only a subset of syllables is given (e.g., in a timed test), it may be necessary to answer additional syllables to obtain a reliable estimate for participants of higher ability levels.

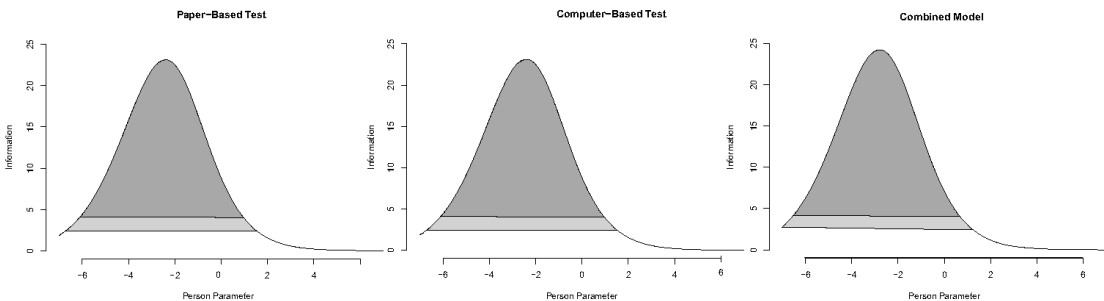

**Figure 1.** Test Information Curves. note: Darkly shaded areas correspond to levels of θ where test reliability is greater than 80%, and lightly shaded areas correspond to levels of θ where test reliability is greater than 70%.

### 3.2. Differences in Sum Scores and Percentage Correct

Paired *t*-tests confirmed that participants answered more syllables (i.e., reading speed) in the paper-based form ($M = 46.4$, $SD = 23.9$) than in the computer-based form ($M = 34.9$, $SD = 17.7$, $t(358) = -19.69$, $p < 0.001$). When excluding not-attempted syllables, the percentage correct (i.e., reading accuracy) did not vary between the paper-based form ($M = 85.2\%$, $SD = 16.3\%$) and the computer-based form ($M = 85.5\%$, $SD = 17.8\%$, $t(310) = 0.44$, $p = 0.66$).

Figure 2 summarizes the proportion correct in boxplots, with separate categories for students with SEN and without. Overall, neither test version was more difficult, but participants answered more quickly in the paper-based version.

### 3.3. Differential Item Functioning

Differential item functioning was not detected for any syllables based on test modality, gender, grade level, migration background, or SEN; all Bonferroni corrected *p* values > 0.05.

### 3.4. Multilevel Modes

Table 2 shows test performance is measured by both sum score and by WLE of theta.

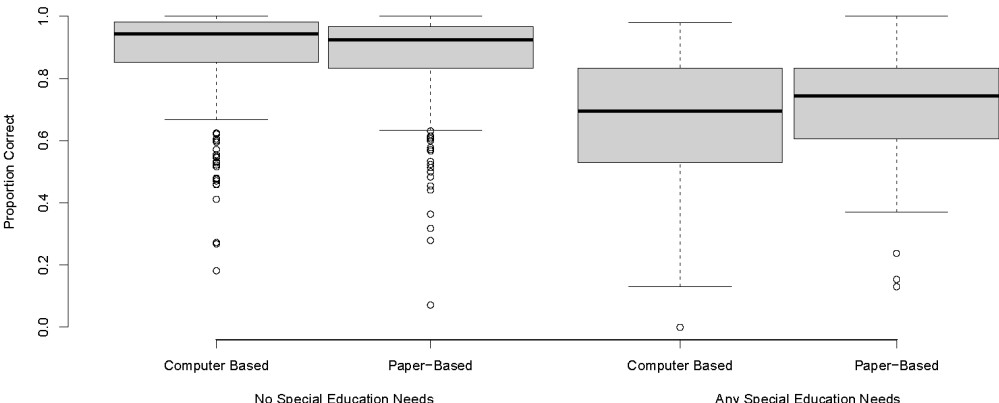

**Figure 2.** Boxplots of proportion correct for computer-based and paper-based tests. note: Not-reached syllables are considered missing and are not included in the proportions.

**Table 2.** Standardized coefficients of multilevel models predicting test performance.

| | Sum Score | | Theta | |
|---|---|---|---|---|
| | Paper-Based $M$ (*SE*) | Computer-Based $M$ (*SE*) | Paper-Based $M$ (*SE*) | Computer-Based $M$ (*SE*) |
| Female | 0.00 (0.06) | 0.00 (0.06) | −0.05 (0.07) | −0.03 (0.06) |
| Migration Background | −0.13 (0.05) ** | −0.12 (0.05) * | −0.15 (0.05) ** | −0.10 (0.06) ˣ |
| Special Education Needs | −0.34 (0.06) *** | −0.38 (0.06) *** | −0.33 (0.06) *** | −0.42 (0.05) *** |
| Third-Year | 0.59 (0.16) *** | 0.45 (1.8) *** | 0.52 (0.20) *** | 0.23 (0.24) |
| R2 within classrooms | 0.14 (0.05) *** | 0.17 (0.05) *** | 0.14 (0.04) ** | 0.20 (0.04) *** |
| R2 within classrooms | 0.35 (0.19) | 0.20 (0.16) *** | 0.27 (0.21) | 0.05 (0.11) |

Note: All values are standardized coefficients. Categories are dummy coded into binary variables. Third-year is a between classrooms comparison of third-year compared to second-year classes. All other comparisons are within classrooms. $R^2$ represents the total variance of the value explained in the model. Sum scores reflect the total number of correctly answered syllables. Theta reflects the WLE of theta. Sum scores and theta values are modeled separately. ˣ $p < 0.10$. * $p < 0.05$. ** $p < 0.01$. *** $p < 0.001$.

Students with a migration background did consistently worse on the test, as did participants with SENs. Third-year students did significantly better than second-year students as measured by sum score on both test versions and as measured by theta estimates in the paper-based form. This difference was not significant for theta-estimates of the computer-based test version.

## 4. Discussion

Our study extends research on mode effects in CBM (i.e., paper-based vs. computer-based test administration) by a focus on ORF measurement because the test-taker completes the test with a teacher involved at the same time. Formative assessments such as CBM are considered a central component of RTI models [9]. Strengthening formative assessment, in particular, is still needed because special education teachers rate their knowledge, experiences, and self-efficacy lower than that related to summative assessment [10,17]. The present study results indicate that both forms are suitable for the measurement of reading aloud and thus support a broader range of test modes in formative assessment. Both test forms show sufficiently good and comparable reliabilities (WLE reliability > 0.76) which is in line with previous results [34]. The Rasch analyses also confirmed that both test forms are suitable for students with rather low ability levels. The used ORF test was constructed as a word list for beginning and struggling readers in line with Fuchs et al. [53] to measure students with low reading levels. The oldest students in the present sample had been attending reading classes in the third grade, so some students were expected to show high reading levels. The test information curves (Figure 1) of both test forms show a similar pattern, namely that for measuring high reading levels additional syllables are required which can be taken as further confirmation of the suitability and usability of both forms.

Students read significantly more syllables correctly in the paper-based version within 60 s while the proportion of correctly and incorrectly solved tasks did not differ between the test forms. Thus, while the sum scores of the two forms cannot be compared without taking into account the differences in the means, the scores for reading accuracy are directly comparable. The lower sum scores can be explained from several perspectives. On the one hand, multiple mode effect studies described that students generally achieve higher sum scores in paper-based CBMs than in computer-based CBMs where each student completes the test on their own (e.g., [53,56,57]). On the other hand, Schaffer Seits [58] showed that students read more words per minute aloud when texts for ORF measures are presented on paper than on screen. Taken together, all results suggest that the computer-based test form reduces processing speed. This could be due to procedural differences involving the presentation of individual syllables on the screen compared to word list presentation on paper. The percentage correct was also expected to be lower on the computer than on the paper-based version [59]. However, we found comparable levels meaning both test modes can be used in the field. The reading task in the study according to Lenhard et al. [59], however, aims at the comprehension level and in our case at the correct decoding. Since the reading tasks in the studies have different demands (reading comprehension vs. correct decoding), further investigations are necessary to find out a possible influence of computer-based testing.

Multilevel modeling showed similar effects of covariates across test modalities. In both test versions, students with SENs and with migration backgrounds read fewer syllables in the time allotted compared to other students, while third graders read more words than second graders. Additionally, students with SENs were less accurate on both test versions. However, third-year students were only more accurate on the paper-based test. This may be because, for students of higher ability, the test was less reliable, but third-grade students were able to complete more problems in the paper-based version, but not sufficient problems in the computer-based version. We also found that while the effect of migration background on ability level was in the predicted direction, it was not statistically significant ($p = 0.098$). More investigation into this effect is likely warranted. However, for everyday classroom purposes, teachers will probably only use the sum scores that show consistent predicted patterns across both test versions.

## 5. Limitations and Future Directions

Our study is limited by several constraints. The ORF test used is constructed as a word list with individual syllables as items and thus is designed especially for beginning readers and students with low reading achievement. Results have shown that both the paper-based and computer-based tests measure very well in the lower ability range of second- and third-graders. We lack an extension of the sample with students in the first grade to see if the comparability of the two test modes can be confirmed in even lower ability ranges. Furthermore, there are ORF forms that work with connected texts, single words, or pseudo words instead of syllables word lists (e.g., [34,53,58]). It remains to be verified whether comparable results can also be replicated in the different types of ORF measures. Another limitation arises from the fact that CBM tests are constructed for repeated use longitudinally. Our study was cross-sectional with only one measurement point. Therefore, future studies should investigate how stable these effects are in repetitions of the tests.

## 6. Conclusions

Our results confirm that there is no differential item functioning between the computer-based and paper-based versions. Both values, sum score and percentage correct, can be used and compared in practice. However, before sum scores of the different test modes can be interpreted jointly, the difference in mean values must be taken into account. In cross-sectional comparisons, both test modes can be used for research and education since they are reliable and valid. Teachers are already familiar with paper-based methods, and there is a higher comparability with other paper-based tests. Computer-based tests offer automatic

scoring and allow for remote administration, which can reduce a teacher's workload. For longitudinal studies, it is not advisable to mix form types because both test forms are reliable but not directly comparable. Especially for school practice, it is recommended to collect and evaluate data with only one test form, since the differences in the sum scores of multiple test modes can vary for each individual child. Comparability in practice could be ensured, for example, by parallel measurement with both test forms at one point in time with subsequent offsetting of the values. However, this requires both a great deal of effort and knowledge of educational assessment data.

The results support the use of both test modes and, as a consequence, teachers should be trained in the use of both computer-based and paper-based tests. This would enable them to use either or both test modes according to classroom demands and personal preferences. Teacher training should include knowledge about the differences between computer-based and paper-based test scores and what variables can influence a test score in order to make informed decisions about test selection. In addition, the use of computer-based testing formats should be covered in teacher training. This could also change teachers' attitudes toward computer-based assessments. Such efforts can reduce teachers' organizational work and safeguard learning and diagnosis in special times such as the COVID-19 pandemic in the long run.

**Author Contributions:** Conceptualization, J.J., J.M.D. and M.G.; methodology, J.M.D.; validation, J.J., J.M.D. and M.G.; formal analysis, J.M.D.; investigation, J.J.; resources, J.M.D.; data curation, J.M.D.; writing—original draft preparation, J.J. and J.M.D.; writing—review and editing, M.G.; visualization, J.J. and J.M.D.; supervision, M.G.; project administration, J.J. and J.M.D. All authors have read and agreed to the published version of the manuscript.

**Funding:** This research received no external funding.

**Institutional Review Board Statement:** All data were collected in the federal state North Rhine-Westphalia, Germany. Permission for this study was granted by the dean of the Faculty of Rehabilitation Science, TU Dortmund University, Germany. Following the requirements of the ministry of education of the federal state North Rhine-Westphalia (Schulgesetz für das Land Nordrhein-Westfalen, 2018), school administrators decided in co-ordination with their teachers on their participation in this scientific study. An additional ethics approval was not required for this study as per the Institution's guidelines and national regulations. Ethic Committee Name: Joint Ethics Committee of the Faculties 12–16, TU Dortmund University Approval Code: 2019-01 Approval Date: 3 June 2019.

**Informed Consent Statement:** Informed consent was obtained from all subjects involved in the study.

**Data Availability Statement:** All Data and Syntax are available under https://osf.io/s4ruc/ (accessed on 12 June 2023).

**Conflicts of Interest:** The authors declare no conflict of interest.

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
