# Peer review of "Measuring Oral Reading Fluency (ORF) Computer-Based and Paper-Based: Examining the Mode Effect in Reading Accuracy and Reading Fluency"

_education, doi:10.3390/educsci13060624_

Round 1

Reviewer 1 Report

1. While you are measuring oral reading fluency, it was never defined. Please define and cite your definition of ORF.

2. The definition of the variables is unclear and confusing. ORF is most often considered as correct-words-per-minute which consists of the indicators of reading rate (or pace) and word identification accuracy. It appears the authors are using the number of correctly pronounced syllables as the measure for rate, but I am not certain. This will be very confusing to many readers who are familiar with the typical indicators of fluency. Please explicitly define the variables and include a rationale for why they deviate from the norm.

3. In line 246 a reference is made to "answered more items correctly" - again, it is unclear as to what items. Please explain.

4. There is no theoretical framework presented for the importance of oral reading fluency and why it is measured. This is necessary to reinforce the importance of the study.

Author Response

Dear Reviewer,

we thank you very much for your comments. They were very helpful, so we happily revised our manuscript and submit it herewith in the adapted form. You will find detailed responses to all comments made. All answers and changed passages that were made in the manuscript are marked. When we refer to specific lines on the manuscript, we refer to the revised version including tracking.

1. While you are measuring oral reading fluency, it was never defined. Please define and cite your definition of ORF.

Answer:

We operationalized ORT tests and defined ORF. Moreover, we added the pedagogical relevance of ORF for reading acquisition.

77f. “In most ORF tests, the number of correctly read syllables or words within 60 seconds is scored to quantify the reading rate in terms of reading speed [27].”

119f. „Reading fluency represents an interplay of accuracy, speed, and prosodic aspects and is understood as the ability to read texts automatically and unconsciously [49]. For early reading learners, it is considered as an important learning step toward becoming a proficient reader [50] because it affects comprehension to a significant degree [51]. Without sufficient reading fluency, readers focus on decoding which limits their ability to comprehend at the same time.“

2. The definition of the variables is unclear and confusing. ORF is most often considered as correct-words-per-minute which consists of the indicators of reading rate (or pace) and word identification accuracy. It appears the authors are using the number of correctly pronounced syllables as the measure for rate, but I am not certain. This will be very confusing to many readers who are familiar with the typical indicators of fluency. Please explicitly define the variables and include a rationale for why they deviate from the norm.

Answer:

We added a detailed description of ORF measures to clarify the differences for the reader. Also, you are correct: in our study, syllables were used as items. This is because the ORF form using syllables does not overwhelm children with low reading skills in particular. Since students with SEN also participated in our study, it was this ORF form that was selected.

124f.: “In ORF tasks, students are presented with either connected texts or lists of un-connected syllables, words, or pseudowords. They are asked to correctly read as many items (i.e., syllables or words) as possible in 60 seconds. The number of items read correctly per minute is often also called the sum score in ORF tasks. Different types of items are used in ORF tasks because the sum score is sensitive to the difficulty level of the items [52]. Unconnected word lists with single syllables or simple words are par-ticularly suitable for measuring the reading performance of students with low skills such as students at the beginning of reading acquisition or those with special education needs [34,53].”

3. In line 246 a reference is made to "answered more items correctly" - again, it is unclear as to what items. Please explain.

Answer:

We have checked the entire manuscript and replaced "items" with "syllables" throughout.

202f.: “All syllables are generated according to defined rules for their respective difficulty level.”

240f.: “The sum score is the total number of correctly answered syllables and reflected reading speed.”

262f.: “A Bonferroni correction was applied based on the number of syllables tested.”

265f.: “After examining the possibility of DIF based on test modality, we examined the number of syllables successfully completed via a within groups comparison. We examined both the number of syllables completed correctly (i.e., sum score) and the IRT estimate of person ability (i.e., theta value).”

282f.: ”Figure 1 shows that when all syllables are answered, the test is generally well targeted for participants of low through moderate ability. Further, the information curves are quite consistent between test modalities. It is important to note that when only a subset of syllables is given (e.g., in a timed test), it may be necessary to answer additional syllables to obtain a reliable estimate for participants of higher ability levels.”

307f.: “Differential item functioning was not detected for any syllables based on test modality, gender, grade level, migration background, or SEN, all Bonferroni corrected p values > 0.05.”

350f.: “Students read significantly more syllables correctly in the paper-based version within 60 seconds while the proportion of correctly and incorrectly solved tasks did not differ between the test forms.“

4. There is no theoretical framework presented for the importance of oral reading fluency and why it is measured. This is necessary to reinforce the importance of the study.

Answer:

First, we integrated the RTI framework in the first section of the instruction to situate the use of ORF tasks in general. Second, we integrated a section on the ORF task into the test mode section and added a statement regarding the importance of reading fluency for both reading acquisition and measurement. The definition of reading fluency is also given in this section.

37: “RTI is a framework in which multiple levels of support are implemented at different intensities to respond to students’ individual learning needs. The main idea is that the greater the student's need, the greater the intensity of the intervention [9]. Typically, RTI models are structured with three tiers. In the first tier, teachers design high-quality in-struction for the entire learning group and assess students’ response to the instruction three times per school year. In the second tier, students who have not sufficiently ben-efited from the first tier’s efforts receive more intensive and preventive interventions in small groups. Brief interventions are usually provided daily and evaluated monthly. At the highest tiers, students with severe difficulties receive intensive and individualized evidence-based interventions, the success of which may be evaluated as often as weekly. To evaluate students’ responses to interventions, classroom-based assessments are implemented at all tiers. Classroom-based assessments include both standardized summative and formative assessments [10]. Especially for students with learning dif-ficulties and SEN in inclusive education, the use of formative assessments (i.e., ’as-sessments for learning‘, [11]) has a positive impact on their learning development (e.g., [12–14]). Formative assessments include repeated measures of specific learning out-comes that allow the process of learning to be visualized [15]. Such assessments are considered crucial components of inclusive education because individual learning needs are expected to be broader than in regular classrooms due to higher heterogeneity in student achievement [16]. Nevertheless, teachers’ familiarity with formative assessments in inclusive education practice is currently still lower than that with summative assessments [10, 17]. A well-known form of formative assessment is curriculum-based measurement (CBM, [18]) which was designed to solve academic difficulties in special education. Multiple instruments have been supported in research as reliable and valid [19–21].”

119f.: „ ORF tasks differ in test administration compared to other CBM tests. They are administered as individual tests because a person with good reading skills is needed to be measured in reading fluency. Reading fluency represents an interplay of accuracy, speed, and prosodic aspects and is understood as the ability to read texts automatically and unconsciously [49]. For early reading learners, it is considered as an important learning step toward becoming a proficient reader [50] because it affects comprehension to a significant degree [51]. Without sufficient reading fluency, readers focus on de-coding which limits their ability to comprehend at the same time. Reading aloud is considered as a robust indicator of reading proficiency [25,27] and is measured via ORF tasks.“

Reviewer 2 Report

The paper looks at an interesting and timely topic. The introduction sets the scene of the study in an effective way, providing a clear account on the statement of the problem, yet contextualising the problem within the specific setting of the study would enrich the introduction even more.

The literature review is all-embracing, current, well discussed and well problematised, leading sensibly to establishing the significance of the foci of this paper. However, the theoretical underpinning of the study is not clear. This part needs to be fully addressed by discussing the theory(ies) that underlie Curriculum-based Measurements. 

In section 2, please abate this claim "To date, there has been no research examining the reading speed (i.e., sum scores) and reading accuracy (i.e., percentage correct) of ORF assessments in which students read computer-based on the screen and teachers use the technology for assessment." Otherwise, please, explain how you have reached it.

The methodology section is well explicated. The selected methods are a good fit for the research objectives. For further improvement, this part needs to provide a detailed account on test validity.   

The discussion is appropriate but can be more insightful and more profound. The first two paragraphs in Section 4 are summative and repetitive. I suggest that you use the word count more effectively, commenting the meaning of the findings and how they link to the theories (once these are added as suggested earlier).  

The paper is well written and well structured, although I would prefer that part 5 be the last part in the paper. 

Author Response

Dear Reviewer,

we thank you very much for your comments. They were very helpful, so we happily revised our manuscript and submit it herewith in the adapted form. The language was carefully reviewed. You will find detailed responses to all comments made. All answers and changed passages that were made in the manuscript are marked. When we refer to specific lines on the manuscript, we refer to the revised version including tracking.

1. The paper looks at an interesting and timely topic. The introduction sets the scene of the study in an effective way, providing a clear account on the statement of the problem, yet contextualising the problem within the specific setting of the study would enrich the introduction even more.

Answer:

Thanks for your feedback. In addition to the supplements on the RTI framework and the educational use of CBM (see the answer to your next comment), we defined ORF in more detail. We also added the pedagogical relevance for reading acquisition.

121f. „ Reading fluency represents an interplay of accuracy, speed, and prosodic aspects and is understood as the ability to read texts automatically and unconsciously [49]. For early reading learners, it is considered as an important learning step toward becoming a proficient reader [50] because it affects comprehension to a significant degree [51]. Without sufficient reading fluency, readers focus on decoding which limits their ability to comprehend at the same time.“

2. The literature review is all-embracing, current, well discussed and well problematised, leading sensibly to establishing the significance of the foci of this paper. However, the theoretical underpinning of the study is not clear. This part needs to be fully addressed by discussing the theory(ies) that underlie Curriculum-based Measurements. 

Answer:

We integrated the RTI framework in the first section of the instruction to situate the use of ORF tasks.

30f.: “Educational Classroom-based assessment data is used in multi-tiered systems of support such as response-to-intervention models (RTI) by teachers to shape teaching and individual learning in the sense of data-based decision-making (DBDM, [1]) with the goal to improve the learning environment for all students in inclusive, special, and regular schools [2–4]. Not all students benefit in the same way from regular instruction (e.g., [5]), and therefore, individualized instructions are needed to meet individual learning needs [6]. In particular, students with special education needs (SEN) are at high risk for difficulties in reading and mathematics [7,8]. RTI is a framework in which mul-tiple levels of support are implemented at different intensities to respond to students’ individual learning needs. The main idea is that the greater the student's need, the greater the intensity of the intervention [9]. Typically, RTI models are structured with three tiers. In the first tier, teachers design high-quality instruction for the entire learning group and assess students’ response to the instruction three times per school year. In the second tier, students who have not sufficiently benefited from the first tier’s efforts receive more intensive and preventive interventions in small groups. Brief in-terventions are usually provided daily and evaluated monthly. At the highest tiers, students with severe difficulties receive intensive and individualized evidence-based interventions, the success of which may be evaluated as often as weekly. To evaluate students’ responses to interventions, classroom-based assessments are implemented at all tiers. Classroom-based assessments include both standardized summative and formative assessments [10]. Especially for students with learning difficulties and SEN in inclusive education, the use of formative assessments (i.e., ’assessments for learning‘, [11]) has a positive impact on their learning development (e.g., [12–14]). Formative as-sessments include repeated measures of specific learning outcomes that allow the process of learning to be visualized [15]. Such assessments are considered crucial components of inclusive education because individual learning needs are expected to be broader than in regular classrooms due to higher heterogeneity in student achievement [16]. Nevertheless, teachers’ familiarity with formative assessments in inclusive edu-cation practice is currently still lower than that with summative assessments [10, 17]. A well-known form of formative assessment is curriculum-based measurement (CBM, [18]) which was designed to solve academic difficulties in special education. Multiple instruments have been supported in research as reliable and valid [19–21].”

3. In section 2, please abate this claim "To date, there has been no research examining the reading speed (i.e., sum scores) and reading accuracy (i.e., percentage correct) of ORF assessments in which students read computer-based on the screen and teachers use the technology for assessment." Otherwise, please, explain how you have reached it.

Answer:

In the manuscript, we reworded the sentence to clarify. However, we would be very pleased if you could provide us with relevant references. We have not found any corresponding papers in a comprehensive search.

168f.: “To dateThe authors could not find any, there has been no research examining the reading speed (i.e., sum scores) and reading accuracy (i.e., percentage correct) of ORF assessments in which students read computer-based on the screen and teachers use the technology for assessment.”

4. The methodology section is well explicated. The selected methods are a good fit for the research objectives. For further improvement, this part needs to provide a detailed account on test validity.   

Answer:

We added correlations between the test used and a standardized and established German-language reading test.

213f.: “In addition, the ORF test used shows moderate to high correlations (r = 0.65—0.71) with the standardized reading comprehension test ELFE II [64] at the word, sentence, and text levels [12].”

5. The discussion is appropriate but can be more insightful and more profound. The first two paragraphs in Section 4 are summative and repetitive. I suggest that you use the word count more effectively, commenting the meaning of the findings and how they link to the theories (once these are added as suggested earlier).  

Answer:

We connected the first two paragraphs via a link back to RTI models and the need to spread formative assessments. In the process, we deleted general summaries.

330f.: “Our study extends research on mode effects in CBM (i.e., paper-based vs. com-puter-based test administration) by a focus on ORF measurement because the test-taker completes the test with a teacher involved at the same time. Formative as-sessments such as CBM are considered a central component of RTI models [9]. Strengthening formative assessment, in particular, is still needed because special edu-cation teachers rate their knowledge, experiences and self-efficacy lower than that re-lated to summative assessment [10,17]. The present study results indicate that both forms are suitable for the measurement of reading aloud and thus support a broader range of test modes in formative assessment. Results indicate that both forms are suitable for the measurement of reading aloud. Both test forms show sufficiently good and comparable reliabilities (WLE reliability > 0.76) which is in line with previous re-sults [34]. The Rasch analyses also confirmed that both test forms are suitable for stu-dents with rather low ability levels. The used ORF test was constructed as a word list for beginning and struggling readers in line with Fuchs et al. [63] to measure students with low reading levels. The oldest students in the present sample had been attending reading classes in third grade, so some students were expected to show high reading levels. The test information curves (Figure 1) of both test forms show a similar pattern namely that for measuring high reading levels additional items syllables are required which can be taken as further confirmation of the suitability and usability of both forms.”

6. The paper is well written and well structured, although I would prefer that part 5 be the last part in the paper. 

Answer:

This order is given by the template of MDPI and is also found in the current contributions. Therefore, we have maintained this order. The paper ends with the section "Conclusions".

Round 2

Reviewer 1 Report

The manuscript is greatly improved from the original version and provides important insights into the use of paper versus digital ORF measures.